# EAGLE: Efficient Analytical Gradient Linear Evaluation for Enhanced Recomputation in Large Language Models

## Abstract

Training large language models requires substantial memory to store intermediate activations, often exceeding the capacity of modern accelerators. Gradient checkpointing addresses this challenge by trading computation for memory, but introduces significant overhead due to forward passes during recomputation. In this work, we present EAGLE (Efficient Analytical Gradient Linear Evaluation), a recomputation strategy that leverages closed-form gradients for linear transformations and integrates FlashAttention's backward algorithm for attention blocks. Unlike traditional checkpointing that uniformly replays the forward pass with autograd, EAGLE computes gradients analytically for linear layers and calls FlashAttention's backward directly, while keeping standard recomputation for nonlinear operations. On production-scale models including DeepSeek-V2, DeepSeek-V3, and LLaMA3-70B (70B–694B parameters), EAGLE improves Model FLOPs Utilization by 18–33% over Full Recompute and achieves up to $9.75\times$ module-level recomputation speedups, and our analysis and experiments show that these gains are achieved without changing training convergence.

## 1 Introduction

The rapid growth of large language models has significantly influenced artificial intelligence, with modern Transformer architectures Vaswani et al. (2017) scaling to large sizes. Contemporary models such as GPT-5 OpenAI (2025), Gemini 2.5 Pro Google DeepMind (2025), Llama 4 Meta AI (2025), and Claude Opus 4.1 Anthropic (2025) require advanced memory management strategies to accommodate their large parameter counts and computational demands. The memory requirements during training encompass multiple components: model parameters, optimizer states Rajbhandari et al. (2020), gradients, and critically, intermediate activations stored for backpropagation Goodfellow et al. (2016).

For large-scale models, activation memory constitutes a substantial portion of the per-device memory footprint during training, creating memory constraints on modern accelerators even with advanced parallelization strategies. As model sizes continue to grow, this memory limitation often restricts the batch sizes and sequence lengths that can be processed efficiently. The growing memory demands have motivated research into memory optimization techniques, ranging from mixed precision training Micikevicius et al. (2018) to advanced parallelization strategies Shoeybi et al. (2019).

A widely adopted approach to address memory constraints is gradient checkpointing, also known as activation recomputation Chen et al. (2016). This technique trades computation for memory by storing only a subset of activations and recomputing the remainder during backpropagation. The basic idea is simple: instead of keeping all intermediate activations in memory, we save only checkpoints at certain layers and recompute the activations between checkpoints when needed during the backward pass. While foundational work established the theoretical framework Chen et al. (2016), subsequent research has explored checkpointing strategies Jain et al. (2020), graph-theoretic approaches Kumar et al. (2019), and specialized techniques for Transformer architectures Korthikanti et al. (2022). However, these methods introduce computational overhead due to their reliance on redundant forward recomputation.

Although recent developments have improved the efficiency of gradient checkpointing Jain et al. (2020); Korthikanti et al. (2022); Zhao et al. (2024); Li et al. (2025), existing methods typically employ a uniform recomputation strategy for all operations, regardless of their mathematical properties. This approach does not exploit the inherent structure of linear layers, which are prevalent throughout Transformer architectures. Specifically, for a linear transformation $\mathbf{y} = \mathbf{W}\mathbf{x}$, the gradients with respect to the weights and inputs admit closed-form solutions: $\nabla_W = \nabla_y \mathbf{x}^\mathsf{T}$ and $\nabla_x = \mathbf{W}^\mathsf{T} \nabla_y$. These gradients can be computed directly using the stored input $\mathbf{x}$ and the incoming gradient $\nabla_y$, without the need to recompute the forward output $\mathbf{y}$. Given that linear transformations are fundamental components in attention projections, MLP modules, and output layers, leveraging their analytical gradients can reduce redundant computation in large-scale model training.

In this work, we present EAGLE (Efficient Analytical Gradient Linear Evaluation), which exploits this mathematical structure to reduce computational overhead in gradient checkpointing. EAGLE applies analytical gradient computation to linear layers and integrates with FlashAttention's backward algorithm for attention mechanisms, while using standard recomputation for nonlinear operations. This selective approach maintains the memory benefits of gradient checkpointing while reducing the computational penalty.

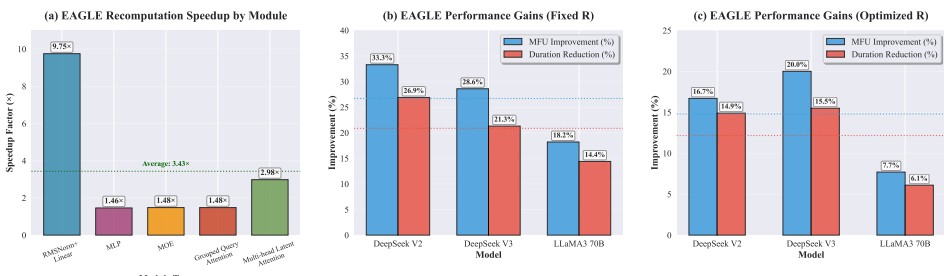

Figure 1: EAGLE achieves consistent improvements from module-level recomputation speedups over the standard autograd-based recomputation (Full Recompute) baseline to end-to-end MFU and throughput gains at similar peak memory.

Figure 1 provides a comprehensive overview of EAGLE's performance across different scales and evaluation settings. The **left panel** shows module-level recomputation speedups for key Transformer components (data from Table 2), with speedup defined relative to the standard autograd-based Full Recompute baseline. The **center panel** presents model-level MFU improvements under fixed recomputation configuration, while the **right panel** shows MFU improvements under optimized recomputation configuration (data from Tables 3 and 4). Detailed peak-memory and performance statistics for No Recompute, Full Recompute, fine-grained recomputation, and EAGLE are reported in Tables 3 and 4. These results validate our analytical gradient approach across diverse architectural components and distributed training configurations.

EAGLE is complementary to existing recomputation and scheduling methods such as fine-grained checkpointing Korthikanti et al. (2022), CheckMate-style tensor rematerialization Jain et al. (2020), and Adacc Li et al. (2025). These approaches determine where and when to recompute activations or how to compress them to save memory, whereas EAGLE focuses on how a chosen recompute region is executed by eliminating redundant forward passes via analytical gradients and FlashAttention integration, so its kernels can be plugged into those frameworks to provide additional execution-level speedups at similar memory.

In practice, EAGLE is most useful when (i) activation memory is the bottleneck so that recomputation is required, and (ii) the recompute path has small inputs, large intermediate activations, and a compute-heavy final layer. This pattern matches standard Transformer blocks such as RMSNorm+Linear projections, MLP up/down projections, MoE experts, and attention blocks ending in a linear output or FlashAttention. In these blocks, EAGLE can be applied on top of existing checkpointing and scheduling strategies to reduce recomputation cost.

Based on these findings, our main contributions are:

- We introduce EAGLE, which uses analytical gradients for linear transformations, achieving recomputation speedups of $9.75\times$ for RMSNorm+Linear, $1.46\times$ for MLP, and $1.48\times$ for MOE modules relative to the standard autograd-based Full Recompute baseline.

- We extend EAGLE to attention mechanisms by integrating with FlashAttention Dao et al. (2022), achieving $1.48\times$ speedup for Grouped Query Attention and $2.98\times$ speedup for Multi-head Latent Attention over the same Full Recompute baseline.

- We demonstrate that on large-scale models (DeepSeek-V2, DeepSeek-V3, LLaMA3-70B; 70B–694B parameters), EAGLE improves MFU over Full Recompute by 18.18%–33.33% (26.69% on average) under fixed recomputation configurations and by 7.69%–20.00% (14.79% on average) under optimized configurations, and both our analysis and appendix experiments indicate that these MFU gains do not come at the cost of changed convergence behavior.

## 2 METHODOLOGY

EAGLE addresses gradient checkpointing's computational overhead by exploiting the mathematical structure of linear transformations. Our key insight is that while gradient checkpointing traditionally recomputes Chen et al. (2016) all forward operations during backpropagation, linear transformations possess closed-form gradient expressions that eliminate the need for forward recomputation. Specifically, for linear layers, gradients can be computed directly from cached inputs and incoming gradients, bypassing the expensive forward pass required by standard recomputation.

We demonstrate EAGLE's effectiveness on two critical Transformer components where linear operations dominate: MLP modules and attention mechanisms, which together constitute the majority of computational cost in modern architectures. Throughout this paper, we use the mathematical notation defined in Appendix B.

### 2.1 OPTIMIZING MLP RECOMPUTATION

---
**Algorithm 1** Standard Recomputation for MLP

---
1: **Forward Pass:**
2:     Input $\mathbf{x}_1$
3:     $\mathbf{y}_1 \leftarrow \mathbf{W}_1\mathbf{x}_1$                                            ▷ Save $\mathbf{x}_1$
4:     $\mathbf{x}_2 \leftarrow \sigma(\mathbf{y}_1)$
5:     $\mathbf{y}_2 \leftarrow \mathbf{W}_2\mathbf{x}_2$
6: **Backward Pass:**
7:     Input $\nabla_{y_2}$
8:     $\mathbf{y}_1 \leftarrow \mathbf{W}_1\mathbf{x}_1$                        ▷ Recompute using saved $\mathbf{x}_1$
9:     $\mathbf{x}_2 \leftarrow \sigma(\mathbf{y}_1)$                                      ▷ Save $\mathbf{y}_1$
10:    $\mathbf{y}_2 \leftarrow \mathbf{W}_2\mathbf{x}_2$                                      ▷ Save $\mathbf{x}_2$
11:    $\nabla_{x_2}, \nabla_{W_2} \leftarrow \text{backward}(\mathbf{y}_2, \nabla_{y_2}, \mathbf{x}_2)$          ▷ Del $\mathbf{y}_2$
12:    $\nabla_{y_1} \leftarrow \text{backward}(\mathbf{x}_2, \nabla_{x_2}, \mathbf{y}_1)$             ▷ Del $\mathbf{x}_2$
13:    $\nabla_{x_1}, \nabla_{W_1} \leftarrow \text{backward}(\mathbf{y}_1, \nabla_{y_1}, \mathbf{x}_1)$          ▷ Del $\mathbf{y}_1$

---

We demonstrate our approach on MLP blocks, which constitute a substantial portion of Transformer computation. In the standard recomputation process (Algorithm 1), both linear transformations ($\mathbf{W}_1\mathbf{x}_1$ and $\mathbf{W}_2\mathbf{x}_2$) are recomputed during the backward pass. Our key observation is that such recomputation is partially redundant. For a linear mapping $\mathbf{y} = \mathbf{W}\mathbf{x}$, the gradients can be derived directly from cached inputs and upstream gradients

$$\nabla_W = \nabla_y \mathbf{x}^\mathsf{T} \tag{1}$$

$$\nabla_x = \mathbf{W}^\mathsf{T} \nabla_y \tag{2}$$

These formulas yield exact gradients at the same cost as a standard backward pass, but avoid the additional forward multiplication required by recomputation. This optimization is particularly valuable in Transformer architectures, where linear layers are extensively used in attention projections

(query, key, value, and output projections), MLP modules, and final output projections. EAGLE applies this principle by eliminating redundant forward computation for the final linear layer in each recompute region, as detailed in Algorithm 2.

---

**Algorithm 2** EAGLE Recomputation for MLP

---

1: **Forward Pass:**
2:    Input $\mathbf{x}_1$
3:    $\mathbf{y}_1 \leftarrow \mathbf{W}_1 \mathbf{x}_1$                                                ▷ Save $\mathbf{x}_1$
4:    $\mathbf{x}_2 \leftarrow \sigma(\mathbf{y}_1)$
5:    $\mathbf{y}_2 \leftarrow \mathbf{W}_2 \mathbf{x}_2$
6: **Backward Pass:**
7:    Input $\nabla_{y_2}$
8:    $\mathbf{y}_1 \leftarrow \mathbf{W}_1 \mathbf{x}_1$                             ▷ Recompute using saved $\mathbf{x}_1$
9:    $\mathbf{x}_2 \leftarrow \sigma(\mathbf{y}_1)$                                     ▷ Save $\mathbf{y}_1$
10:   **Skip forward computation for $\mathbf{y}_2$**
11:   $\nabla_{x_2} \leftarrow \mathbf{W}_2^\mathsf{T} \nabla_{y_2}$                      ▷ Analytical: $\nabla_x = W^T \nabla_y$
12:   $\nabla_{W_2} \leftarrow \nabla_{y_2} \mathbf{x}_2^\mathsf{T}$                      ▷ Analytical: $\nabla_W = \nabla_y x^T$
13:   $\nabla_{y_1} \leftarrow \text{backward}(\mathbf{x}_2, \nabla_{x_2}, \mathbf{y}_1)$             ▷ Del $\mathbf{x}_2$
14:   $\nabla_{x_1}, \nabla_{W_1} \leftarrow \text{backward}(\mathbf{y}_1, \nabla_{y_1}, \mathbf{x}_1)$       ▷ Del $\mathbf{y}_1$

---

The difference between Algorithm 1 and Algorithm 2 lies in steps 11–12 of the EAGLE version: instead of recomputing $\mathbf{y}_2 = \mathbf{W}_2 \mathbf{x}_2$, we compute $\nabla W_2$ and $\nabla x_2$ directly using Equations 1–2. This saves one matrix multiplication per recompute while keeping the backward pass mathematically identical to standard automatic differentiation for this block. Since these gradients are computed from the same loss, inputs, and upstream gradients as in Autograd, they should produce the same updates in theory. Our gradient check in Appendix E (Table 8) shows that the differences between EAGLE and FP32 Autograd are tiny under default non-deterministic kernels and drop to exactly zero once deterministic kernels are enabled. The convergence and run-to-run variance results (Tables 9 and 10) show discrepancies on the same order as natural run-to-run variability, indicating that EAGLE does not alter training behavior compared to standard recomputation.

## 2.2 Optimizing Attention Recomputation

Attention mechanisms require different optimization approaches compared to linear layers. While the output projection can benefit from analytical gradients as demonstrated in the MLP case, the attention computation requires both QKV projection and expensive core attention forward passes during gradient checkpointing. EAGLE addresses this by combining analytical gradients for the linear output projection with direct integration into FlashAttention's backward algorithm, eliminating redundant computation.

To understand the computational overhead in standard approaches, we first examine the recomputation process for self-attention, as shown in Algorithm 3. In standard recomputation, the backward pass requires a FlashAttention forward pass (step 10) to recompute the attention output before computing attention gradients.

The key insight is that we can eliminate this redundancy. FlashAttention's backward algorithm can compute attention gradients directly without requiring the forward recomputation. Based on this analysis, EAGLE's improved approach is shown in Algorithm 4.

Step 11 directly invokes FlashAttention's backward algorithm using the preserved output $o$, the gradient $\nabla o$ from the output projection backward pass, and the recomputed inputs $q, k, v$. This approach eliminates the attention forward recomputation required in standard methods, directly computing $\nabla q, \nabla k, \nabla v$ while maintaining identical gradient accuracy.

EAGLE maintains the same memory footprint as standard recomputation since the attention output $o$ is preserved in both approaches. The method ensures mathematically equivalent results to automatic differentiation.

---

**Algorithm 3** Standard Recomputation for Attention

1: **Forward Pass:**
2:     Input $x_1$
3:     $q, k, v \leftarrow W_q x_1, W_k x_1, W_v x_1$
4:     $o \leftarrow \text{FA}(q, k, v)$
5:     $y \leftarrow W_o o$
6: **Backward Pass:**
7:     Input $\nabla y$
8:     $\nabla o, \nabla W_o \leftarrow \text{backward}(y, \nabla y, o)$          ▷ Del $o$
9:     $q, k, v \leftarrow W_q x_1, W_k x_1, W_v x_1$
10:    $o \leftarrow \text{FA}(q, k, v)$
11:    $\nabla q, \nabla k, \nabla v \leftarrow \text{FA\_backward}(o, \nabla o, q, k, v)$          ▷ Del $o$
12:    $\nabla x_1, \nabla W_q, \nabla W_k, \nabla W_v \leftarrow \text{backward}(q, k, v, \nabla q, \nabla k, \nabla v, x_1)$          ▷ Del $q, k, v$

---

**Algorithm 4** EAGLE Recomputation for Attention

1: **Forward Pass:**
2:     Input $x_1$
3:     $q, k, v \leftarrow W_q x_1, W_k x_1, W_v x_1$          ▷ Save $x_1$
4:     $o \leftarrow \text{FA}(q, k, v)$
5:     $y \leftarrow W_o o$          ▷ Save $o$
6: **Backward Pass:**
7:     Input $\nabla y$
8:     $\nabla o, \nabla W_o \leftarrow \text{backward}(y, \nabla y, o)$          ▷ Keep $o$
9:     $q, k, v \leftarrow W_q x_1, W_k x_1, W_v x_1$
10:    **Skip forward computation for** $o$
11:    $\nabla q, \nabla k, \nabla v \leftarrow \text{FA\_backward}(o, \nabla o, q, k, v)$          ▷ Direct FA backward
12:    $\nabla x_1, \nabla W_q, \nabla W_k, \nabla W_v \leftarrow \text{backward}(q, k, v, \nabla q, \nabla k, \nabla v, x_1)$          ▷ Del $q, k, v$

---

## 3 EXPERIMENTS

We conduct experiments to evaluate EAGLE's performance across multiple dimensions, from individual module performance to end-to-end training efficiency on large-scale models. Our evaluation includes micro-benchmarks on key Transformer components and macro-benchmarks on production-scale language models ranging from 70B to 694B parameters, including LLaMA3-70B, DeepSeek-V2, and DeepSeek-V3.

A critical hyperparameter in gradient checkpointing is the number of layers per pipeline stage that undergo recomputation, which we denote as $R$. This parameter directly controls the memory-computation trade-off: higher $R$ values reduce memory consumption but increase computational overhead. In our experiments, we explore two evaluation scenarios: fixed $R$ across all methods for fair algorithmic comparison, and optimized $R$ values that maximize Model FLOPs Utilization (MFU) for each method individually. The evaluation shows EAGLE's performance in the memory-computation trade-off by achieving memory efficiency identical to fine-grained recomputation while delivering improved computational performance compared to standard recomputation.

### 3.1 EXPERIMENTAL SETUP

**Recomputation Granularity Design.** Following Korthikanti et al. (2022), we adopt fine-grained recomputation rather than full-layer recomputation to enable flexible trade-offs between computational overhead and memory usage. Our recomputation strategy is tailored to modern Transformer architectures, with specific granularity designs for RMSNorm+Linear projections, Multi-Layer Perceptron (MLP), Mixture of Experts (MOE), Grouped-Query Attention (GQA), and Multi-head Latent Attention (MLA) modules. Detailed recomputation block partitioning for each module type is provided in Appendix C.

**Baseline Comparison.** Our evaluation compares four distinct recomputation strategies: (1) *No Recompute* stores all activations, serving as an upper bound for computational efficiency while repre-

senting maximum memory consumption; (2) *Full Recompute* implements complete gradient checkpointing at layer boundaries, providing maximum memory savings at higher computational cost; (3) *Fine-grained Recompute* applies recomputation selectively to specific components within each Transformer layer, following established practices to offer balanced memory-computation trade-offs; and (4) *EAGLE* represents our proposed approach, utilizing closed-form gradients for linear layers while maintaining standard recomputation for nonlinear operations.

**Hardware and Training Configuration.** All experiments are conducted on PCIe A100 GPUs with 80GB memory. For large-scale model evaluation, DeepSeek-V2 and DeepSeek-V3 experiments utilize 2048 GPUs, while LLaMA3-70B experiments are conducted on 32 GPUs. The distributed training employs mixed precision (BF16) with detailed model parameters and parallelization strategies provided in Appendix D.

**Performance Metrics.** For module-level analysis, we measure execution time and memory consumption to evaluate computational efficiency of individual components. For model-level evaluation, we employ MFU, training iteration duration, and peak memory consumption per device across pipeline stages.

Table 1: Module-level memory consumption and computational complexity comparison across recomputation strategies.

| Module Type | Memory Usage | FLOPs (F+B) |
|---|:---:|:---:|
| *No Recomputation* | | |
| RMSNorm+Linear | $4bsh$ | $6bshh_1$ |
| MLP | $4bsh + 6bsh_1$ | $18bshh_1$ |
| MOE | $2bs(2h + 3h_1 + Kh_1 + 3Kh_1)$ | $18bshh_1(1 + K)$ |
| Grouped Query Attention | $8bsh + 4bsgd$ | $24bsh^2 + 12bs^2h$ |
| Multi-head Latent Attention | $2bs(2h + \tilde{h} + 4ad + 2ar)$ | $6bs(h\tilde{h} + a(d + r)(h_q + s)$ $+ad(2h_{kv} + s + h))$ |
| *Standard Recomputation* | | |
| RMSNorm+Linear | $2bsh$ | $8bshh_1$ |
| MLP | $2bsh$ | $24bshh_1$ |
| MOE | $2bsh(1 + K)$ | $24bshh_1(1 + K)$ |
| Grouped Query Attention | $4bsh$ | $30bsh^2 + 16bs^2h$ |
| Multi-head Latent Attention | $2bs(h + \tilde{h} + ad)$ | $2bs(4h\tilde{h} + a(d + r)(4h_q + 4s)$ $+ad(8h_{kv} + 4s + 3h))$ |
| *EAGLE* | | |
| RMSNorm+Linear | $2bsh$ | $6bshh_1$ |
| MLP | $2bsh$ | $22bshh_1$ |
| MOE | $2bsh(1 + K)$ | $22bshh_1(1 + K)$ |
| Grouped Query Attention | $4bsh$ | $30bsh^2 + 12bs^2h$ |
| Multi-head Latent Attention | $2bs(h + \tilde{h} + ad)$ | $2bs(3h\tilde{h} + a(d + r)(4h_q + 3s)$ $+ad(8h_{kv} + 3s + 3h))$ |

$\tilde{h} = h_q + h_{kv} + r$ represents the combined latent dimension and decoupled dimension for RoPE.

### 3.2 MODULE-LEVEL PERFORMANCE ANALYSIS

Individual module performance analysis provides insights into EAGLE's computational benefits across different components. We examine the performance characteristics of key module types commonly found in large language models using standardized experimental conditions of batch size $b = 1$ and sequence length $s = 4096$. Tables 1 and 2 demonstrate EAGLE's advantages across three dimensions: memory efficiency, computational reduction, and execution time improvement.

For timing analysis, speedup is calculated as:

$$\text{Speedup} = \frac{\text{Standard Recompute Time}}{\text{EAGLE Recompute Time}} \tag{3}$$

Table 2: Module-level execution time comparison across recomputation strategies.

| Module Type | Forward (ms) | Backward (ms) | Recompute (ms) | Total (ms) | Speedup (Recompute) |
|---|---|---|---|---|---|
| *No Recomputation* | | | | | |
| RMSNorm+Linear | 1.17 | 2.40 | 0.00 | 3.57 | – |
| MLP | 1.81 | 3.63 | 0.00 | 5.44 | – |
| MOE | 62.63 | 95.62 | 0.00 | 158.25 | – |
| Grouped Query Attention | 7.00 | 15.48 | 0.00 | 22.48 | – |
| Multi-head Latent Attention | 11.28 | 27.82 | 0.00 | 39.10 | – |
| *Standard Recomputation* | | | | | |
| RMSNorm+Linear | 1.17 | 2.40 | 1.17 | 4.74 | – |
| MLP | 1.81 | 3.63 | 1.81 | 7.25 | – |
| MOE | 62.63 | 95.62 | 7.54 | 165.79 | – |
| Grouped Query Attention | 7.00 | 15.48 | 4.56 | 27.04 | – |
| Multi-head Latent Attention | 11.28 | 27.82 | 7.07 | 46.17 | – |
| *EAGLE* | | | | | |
| RMSNorm+Linear | 1.17 | 2.40 | 0.12 | 3.69 | $9.75\times$ |
| MLP | 1.81 | 3.63 | 1.24 | 6.68 | $1.46\times$ |
| MOE | 62.63 | 95.62 | 5.11 | 163.36 | $1.48\times$ |
| Grouped Query Attention | 7.00 | 15.48 | 3.08 | 25.56 | $1.48\times$ |
| Multi-head Latent Attention | 11.28 | 27.82 | 2.37 | 41.47 | $2.98\times$ |

**Memory Efficiency Analysis (Table 1):** EAGLE maintains the same memory efficiency as standard recomputation while reducing memory consumption compared to the no-recomputation baseline. RMSNorm+Linear modules achieve 50.00% memory reduction from $4bsh$ (no recomputation) to $2bsh$ (EAGLE). MLP modules show memory reduction, decreasing from $4bsh + 6bsh_1$ to $2bsh$. MOE modules achieve memory reduction from $2bs(2h + 3h_1 + Kh_1 + 3Kh_1)$ to $2bsh(1 + K)$. Multi-head Latent Attention maintains efficient memory usage at $2bs(h+\tilde{h}+ad)$, matching standard recomputation performance.

**Computational Efficiency Analysis (Table 1):** EAGLE reduces FLOPs compared to standard recomputation by eliminating redundant forward computations. RMSNorm+Linear modules achieve 25.00% FLOP reduction (from $8bshh_1$ to $6bshh_1$), while MLP modules achieve 8.33% FLOP reduction (from $24bshh_1$ to $22bshh_1$). MOE modules achieve 8.33% FLOP reduction (from $24bshh_1(1+K)$ to $22bshh_1(1+K)$). For attention mechanisms, EAGLE achieves FLOP reductions compared to standard recomputation while maintaining competitive memory efficiency.

**Execution Time Performance (Table 2):** The timing analysis reveals recomputation speedups across all module types. RMSNorm+Linear modules achieve the largest speedup with $9.75\times$. MLP modules deliver $1.46\times$ speedup, while MOE modules achieve $1.48\times$ speedup. Attention mechanisms show improvements: Grouped Query Attention ($1.48\times$) and Multi-head Latent Attention ($2.98\times$). The variation in speedup factors reflects different linear-to-nonlinear computation ratios across module types.

These advantages, including memory efficiency matching standard recomputation, computational reduction compared to standard recomputation, and execution time improvement across all module types, make EAGLE suitable for memory-constrained large-scale model training.

### 3.3 END-TO-END TRAINING PERFORMANCE EVALUATION

End-to-end training performance evaluation assesses EAGLE's effectiveness in production-scale scenarios across models ranging from 70B to 694B parameters. Our evaluation encompasses three representative architectures with different scales and parallelization strategies, as detailed in the Appendix D.

We analyze memory consumption across three pipeline stages for all models. The **First Stage** contains embedding layers and initial Transformer blocks, the **Middle Stage** represents intermediate

Transformer blocks with identical structure, and the **Last Stage** includes final Transformer blocks and the language modeling head. This stage-wise analysis allows us to understand how EAGLE's memory optimization scales across different pipeline configurations and architectural components.

The DeepSeek-V2 and DeepSeek-V3 models utilize mixture-of-experts architectures with pipeline parallelism, while the LLaMA3-70B model provides comparison with standard dense Transformer architectures using tensor parallelism. This evaluation shows EAGLE's performance across different distributed training paradigms.

In the following performance analysis, we calculate improvements using:

$$\text{Change}(X) = \frac{X_{\text{EAGLE}} - X_{\text{Full Recompute}}}{X_{\text{Full Recompute}}} \times 100\%, \quad X \in \{\text{MFU}, \text{Tokens/s}, \text{Duration}\}. \quad (4)$$

Table 3: Performance comparison with fixed recomputation configuration. All models use the same number of recomputed layers per pipeline stage ($R$) for fair comparison across different recomputation strategies.

| Model & Method | MFU | Duration (s) | Tokens/s | Peak Mem (GB) | First Stage (GB) | Middle Stage (GB) | Last Stage (GB) |
|---|---|---|---|---|---|---|---|
| **DeepSeek-V2** | | | | | | | |
| No Recompute | - | - | – | > 80 | 100.99 | 99.16 | 21.78 |
| Full Recompute | 0.15 | 137.26 | 238.73 | 28.90 | 21.35 | 20.39 | 17.77 |
| Fine-grained Recompute | 0.20 | 104.32 | 314.11 | 54.55 | 43.93 | 45.12 | 19.03 |
| EAGLE | **0.20** | **100.35** | **326.53** | **54.50** | 43.93 | 45.12 | 19.03 |
| *EAGLE vs. Full Recompute* | **+33.33%** | **-26.89%** | **+36.78%** | - | - | - | - |
| **DeepSeek-V3** | | | | | | | |
| No Recompute | - | - | – | > 80 | 110.12 | 121.64 | 23.62 |
| Full Recompute | 0.28 | 108.16 | 302.96 | 32.19 | 24.30 | 18.79 | 18.40 |
| Fine-grained Recompute | 0.34 | 90.55 | 361.88 | 66.13 | 42.66 | 55.96 | 20.32 |
| EAGLE | **0.36** | **85.10** | **385.05** | **66.28** | 42.66 | 55.96 | 20.32 |
| *EAGLE vs. Full Recompute* | **+28.57%** | **-21.32%** | **+27.11%** | - | - | - | - |
| **LLaMA3-70B** | | | | | | | |
| No Recompute | - | - | – | > 80 | 80.22 | 67.54 | 50.09 |
| Full Recompute | 0.11 | 431.66 | 75.91 | 49.10 | 41.43 | 38.59 | 40.24 |
| Fine-grained Recompute | 0.12 | 403.34 | 81.24 | 54.77 | 46.30 | 42.21 | 41.49 |
| EAGLE | **0.13** | **369.63** | **88.65** | **54.86** | 46.30 | 42.21 | 41.49 |
| *EAGLE vs. Full Recompute* | **+18.18%** | **-14.37%** | **+16.78%** | - | - | - | - |
| *Average EAGLE Improvement* | **+26.69%** | **-20.86%** | **+26.89%** | - | - | - | - |

**Fixed Configuration Comparison (Table 3):** We maintain consistent recomputation settings across all methods by using identical values of $R$ to ensure fair algorithmic comparison. Under this fixed configuration, EAGLE improves MFU over Full Recompute by 33.33%, 28.57%, and 18.18% on DeepSeek-V2, DeepSeek-V3, and LLaMA3-70B, yielding a 26.69% average MFU gain while also reducing training duration by 20.86%.

**Optimized Configuration Comparison (Table 4):** We further allow each method to choose its optimal number of recomputed layers $R$ per pipeline stage to maximize MFU, providing a best-case comparison under tuned settings. Even in this scenario, EAGLE maintains clear advantages: it improves MFU over Full Recompute by 16.67% on DeepSeek-V2, 20.00% on DeepSeek-V3, and 7.69% on LLaMA3-70B, yielding a 14.79% average MFU gain, indicating that the benefits come from algorithmic efficiency rather than from more aggressive hyperparameter tuning.

Overall, the experimental results demonstrate EAGLE's effectiveness across different model architectures and scales. Comparing the fixed and optimized configurations shows that EAGLE consistently improves MFU and training time across heterogeneous parallelization strategies, including pipeline parallelism with expert parallelism for DeepSeek models and tensor parallelism for LLaMA3-70B, supporting its usefulness in realistic large-scale distributed training environments.

### 3.4 SUMMARY

Our comprehensive evaluation demonstrates EAGLE's effectiveness across multiple dimensions. At the module level, EAGLE achieves recomputation speedups ranging from $1.46\times$ to $9.75\times$ for com-

Table 4: Performance comparison with optimized recomputation configuration. Each model uses its optimal number of recomputed layers per pipeline stage ($R$) to achieve maximum MFU.

| Model & Method | MFU | Duration (s) | Tokens/s | Peak Mem (GB) | First Stage (GB) | Middle Stage (GB) | Last Stage (GB) |
|---|---|---|---|---|---|---|---|
| **DeepSeek-V2** | | | | | | | |
| No Recompute | - | - | – | > 80 | 100.99 | 99.16 | 21.78 |
| Full Recompute ($R = 2$) | 0.18 | 116.71 | 280.76 | 67.00 | 62.74 | 59.11 | 19.11 |
| Fine-grained Recompute ($R = 3$) | 0.20 | 102.27 | 320.41 | 62.19 | 57.70 | 57.98 | 19.03 |
| EAGLE ($R = 3$) | **0.21** | **99.29** | **330.02** | **62.43** | 57.70 | 57.98 | 19.03 |
| *EAGLE vs. Full Recompute* | **+16.67%** | **-14.93%** | **+17.6%** | - | - | - | - |
| **DeepSeek-V3** | | | | | | | |
| No Recompute | - | - | – | > 80 | 110.12 | 121.64 | 23.62 |
| Full Recompute ($R = 3$) | 0.30 | 100.71 | 325.37 | 54.55 | 50.45 | 43.20 | 18.40 |
| Fine-grained Recompute ($R = 4$) | 0.34 | 90.55 | 361.88 | 60.28 | 42.66 | 55.96 | 20.32 |
| EAGLE ($R = 4$) | **0.36** | **85.10** | **385.05** | **59.90** | 42.66 | 55.96 | 20.32 |
| *EAGLE vs. Full Recompute* | **+20.00%** | **-15.50%** | **+18.3%** | - | - | - | - |
| **LLaMA3-70B** | | | | | | | |
| No Recompute | - | - | – | > 80 | 80.22 | 67.54 | 50.09 |
| Full Recompute ($R = 8$) | 0.13 | 360.11 | 90.99 | 69.23 | 64.47 | 59.27 | 49.10 |
| Fine-grained Recompute ($R = 9$) | 0.13 | 353.33 | 92.74 | 69.54 | 64.75 | 55.93 | 46.22 |
| EAGLE ($R = 9$) | **0.14** | **338.16** | **96.90** | **69.67** | 64.75 | 55.93 | 46.22 |
| *EAGLE vs. Full Recompute* | **+7.69%** | **-6.10%** | **+6.5%** | - | - | - | - |
| *Average EAGLE vs. Full Recompute* | **+14.79%** | **-12.18%** | **+14.1%** | - | - | - | - |

mon LLM building blocks (RMSNorm+Linear, MLP, MOE, and attention blocks). At the model level, EAGLE outperforms existing recomputation strategies across all evaluated architectures: under fixed configurations, it achieves an average 26.69% MFU improvement and 20.86% training time reduction compared to Full Recompute, and even under optimized configurations where baselines operate at their best settings, it maintains a 14.79% average MFU advantage.

The results demonstrate that EAGLE improves the memory-computation trade-off in gradient checkpointing by applying analytical gradients to linear operations while maintaining standard recomputation for nonlinear components. This approach reduces computational overhead compared to standard recomputation while preserving memory efficiency, providing benefits for large-scale model training.

## 4 CONCLUSION

We have presented EAGLE, a gradient checkpointing approach that leverages analytical gradients for linear transformations and integrates with FlashAttention's backward algorithm. EAGLE eliminates redundant forward passes by computing gradients directly from cached inputs for linear operations and combining analytical gradients with direct FlashAttention backward computation for attention mechanisms.

Our evaluation demonstrates consistent improvements across scales: $9.75\times$ speedup for RMSNorm+Linear modules, $1.46\times$ for MLP blocks, and up to $2.98\times$ for attention mechanisms, which are standard building blocks in modern LLM architectures. These optimizations translate to end-to-end MFU improvements of roughly 18%–33% under fixed-layer configurations and 8%–20% under optimized configurations compared to Full Recompute on models ranging from 70B to 694B parameters, and both our analysis and the experiments in Appendix E indicate that these gains do not change convergence behavior.

EAGLE demonstrates that selectively exploiting mathematical structure in gradient checkpointing can achieve computational performance improvements without compromising memory efficiency, making it suitable for large-scale model training where both memory and computation are constrained.

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

## A RELATED WORK

The challenge of memory-efficient training in large language models has driven extensive research into activation recomputation and gradient checkpointing techniques. Chen et al. Chen et al. (2016) established gradient checkpointing as a fundamental approach for trading computation for memory, enabling the training of deeper networks by selectively storing and recomputing activations during backpropagation. This foundational technique has been refined through optimal checkpointing strategies Jain et al. (2020), graph-theoretic frameworks for memory-efficient backpropagation Kumar et al. (2019), and specialized techniques for large Transformer models Korthikanti et al. (2022). However, these approaches uniformly rely on autograd-based recomputation, treating all neural network operations identically without exploiting their underlying mathematical structure.

The scaling of models to trillion parameters has necessitated increasingly sophisticated memory management strategies. ZeRO Rajbhandari et al. (2020) improved distributed training by partitioning optimizer states, gradients, and parameters across devices, while dynamic memory management systems Wang et al. (2022); Huang et al. (2020) provide adaptive allocation policies that respond to runtime memory pressure. Modern frameworks like Megatron-LM Shoeybi et al. (2019) have demonstrated the effectiveness of model parallelism at scale, yet these systems still depend on uniform recomputation strategies that do not differentiate between linear and nonlinear operations. Parameter-efficient approaches like QLoRA Dettmers et al. (2023) have reduced fine-tuning memory requirements through quantization and low-rank adaptation, but the fundamental recomputation paradigm remains unchanged.

Attention mechanisms, with their quadratic memory complexity, have been a primary target for optimization efforts directly relevant to our work. FlashAttention Dao et al. (2022) and its successor FlashAttention-2 Dao (2023) have achieved memory and computational improvements through IO-aware computation and optimized parallelism. These methods demonstrate that mathematical structure can be exploited for efficiency gains—FlashAttention avoids materializing the full attention matrix by leveraging the mathematical properties of softmax and matrix multiplication. Ring Attention Liu et al. (2023) extends this principle to enable extended context processing through distributed computation, while recent compression-based approaches Li et al. (2024) and novel memory allocation strategies Zhang et al. (2023) have addressed memory management challenges in deep learning.

Recent developments have introduced various approaches to memory optimization, yet they continue to overlook the mathematical structure of linear transformations. Adacc Li et al. (2025) presents adaptive compression and activation checkpointing techniques that achieve $1.01 - 1.37\times$ training speedup while maintaining model precision, improving upon traditional checkpointing methods. However, Adacc still applies uniform compression strategies without distinguishing between linear and nonlinear components. MAS-Attention Zhao et al. (2024) introduces memory-aware stream processing for attention acceleration, achieving $2.75\times$ speedup and 54.00% energy reduction, but focuses primarily on hardware optimization rather than mathematical structure exploitation. DeepSeek's Native Sparse Attention DeepSeek-AI (2025) reduces attention complexity from $O(s^2)$ to $O(s \log s)$ through dynamic hierarchical pruning, achieving $11\times$ inference speedup on 128k-length tasks while maintaining 99.30% accuracy, demonstrating the potential of mathematically-informed optimizations.

The evolution of alternative architectures has continued with Mamba2 Dao & Gu (2024), which unifies Transformers and structured state space models through mathematical duality principles, alongside specialized variants like MambaByte Wang et al. (2024a) and MoE-Mamba Pióro et al. (2024). These developments highlight the importance of mathematical structure in achieving computational efficiency. Memory-enhanced approaches like MemLong Wang et al. (2024b) have addressed long-text modeling through external retrieval mechanisms, while continual learning frameworks Chen et al. (2024) have tackled catastrophic forgetting with memory-efficient architectures. Recent distributed systems research has focused on disaggregated architectures, with DistServe Zhong et al. (2024) and Llumnix Sun et al. (2024) optimizing serving efficiency, yet these systems still rely on conventional recomputation strategies.

Despite these advances, existing approaches share a fundamental limitation: they treat all neural network components uniformly, applying autograd-based recomputation without regard for underlying mathematical properties. This is particularly inefficient for linear transformations, which possess

closed-form gradient expressions ($\nabla W = \nabla y \cdot x^T$, $\nabla x = W^T \cdot \nabla y$) that can be computed directly from cached inputs and incoming gradients. Linear layers constitute a substantial portion of Transformer architectures—including attention projections, and MLP components—yet current recomputation strategies redundantly recompute their forward passes. While recent work has successfully exploited mathematical structure in attention mechanisms, no prior research has systematically applied this principle to linear transformations in gradient checkpointing, representing the key opportunity that EAGLE addresses.

## B  NOTATION

This section provides a comprehensive overview of the mathematical notation and variable definitions used throughout this paper. The notation covers model architecture parameters, training configurations, and specialized dimensions for different attention mechanisms.

Table 5: Mathematical notation and variable definitions.

| Symbol | Definition |
|---|---|
| $b$ | Batch size (number of sequences processed together) |
| $s$ | Sequence length (number of tokens per sequence) |
| $h$ | Hidden dimension size of the model |
| $d$ | Attention head dimension |
| $L$ | Number of Transformer layers |
| $n$ | Number of attention heads |
| $t$ | Tensor parallel size |
| $p$ | Pipeline parallel size |
| $v$ | Vocabulary size |
| $h_1$ | Intermediate dimension in MLP |
| $g$ | Number of groups in Grouped Query Attention |
| $a$ | Number of attention heads in Multi-head Latent Attention |
| $r$ | Per-head dimension of the decoupled queries and key to apply RoPE in Multi-head Latent Attention |
| $h_q$ | Compressed latent dimension for Query in Multi-head Latent Attention |
| $h_{kv}$ | Compressed latent dimension for Key-value in Multi-head Latent Attention |
| $\tilde{h}$ | Combined dimension: $h_q + h_{kv} + r$ |
| $K$ | Top K in MOE |
| $R$ | Number of recomputed layers per pipeline stage |

## C  RECOMPUTATION GRANULARITY DETAILS

This section provides detailed specifications for the recomputation granularity design applied to different module types in modern Transformer architectures.

**RMSNorm+Linear Projections** are treated as unified recomputation blocks that encompass the RMSNorm operation followed by linear transformations. This includes attention input projections, and language modeling head projections. The recomputation block preserves the input activations and recomputes the normalization and linear transformation operations during the backward pass.

**Multi-Layer Perceptron (MLP)** applies recomputation to a unified block containing the input RMSNorm, both linear projections (up and down), and the SiLU activation function. The block boundary is designed to minimize memory overhead while maintaining computational efficiency for the feedforward operations.

**Mixture of Experts (MOE)** is divided into two recomputation blocks. The first block recomputes the input RMSNorm, the router projection, and the shared experts. The second block recomputes the routed experts, excluding the all-to-all communication and tensor permutation operations.

**Grouped-Query Attention (GQA)** applies recomputation as a single unified block encompassing the input RMSNorm, all linear projections (query, key, and value), and the FlashAttention operations. This design treats the entire attention mechanism as a cohesive computational unit for recomputation purposes.

**Multi-head Latent Attention (MLA)** is partitioned into two recomputation blocks to optimize the memory-computation trade-off:

- *Block 1:* Recomputes the input RMSNorm and the two subsequent down projections that transform the input to query and key-value representations.
- *Block 2:* Recomputes the latent RMSNorm, the subsequent up projection that expands the latent representation, and the FlashAttention component.

This two-block partitioning respects the natural computational flow of MLA while enabling flexible memory management.

This granularity design balances computational overhead with memory savings while respecting the natural computational boundaries of each module type.

# D   MODEL CONFIGURATIONS

Our evaluation encompasses three representative large language models with diverse architectures to validate EAGLE's effectiveness across different computational paradigms. The model selection includes both dense and mixture-of-experts (MoE) architectures.

LLaMA3-70B represents a standard dense Transformer architecture with grouped-query attention (GQA). The DeepSeek models employ mixture-of-experts architectures with multi-head latent attention (MLA), reflecting the scaling trends in modern sparse architectures.

This architectural diversity allows us to assess EAGLE's performance across different attention mechanisms (GQA vs. MLA) and computational patterns (dense vs. sparse expert activation), demonstrating the generalizability of our analytical gradient approach across contemporary large language model architectures.

Table 6: Model configurations used in our evaluation.

| Model | Parameters | Layers | Hidden | Heads | KV Heads | Experts | TopK | mbs | TP | PP |
|---|---|---|---|---|---|---|---|---|---|---|
| LLaMA3-70B | 70B | 80 | 8192 | 64 | 8 | - | - | 4 | 8 | 4 |
| DeepSeek-V2 | 247B | 63 | 5120 | 128 | 128 | 160 | 6 | 1 | - | 16 |
| DeepSeek-V3 | 694B | 63 | 7168 | 128 | 128 | 256 | 8 | 1 | - | 16 |

# E   ADDITIONAL EXPERIMENTAL ANALYSES

## E.1   SEQUENCE LENGTH SENSITIVITY ON LLAMA3-8B

To complement our large-scale benchmarks, we conduct a sensitivity study on LLaMA3-8B to analyze how EAGLE's efficiency varies with sequence length. We sweep sequence lengths $s \in \{4096, 8192, 16384, 32768\}$ under identical training configurations and compare No Recompute, Full Recompute, selective fine-grained recomputation, and EAGLE. On this mid-scale setup, all configurations, including No Recompute, fit within GPU memory, so the results primarily reflect recomputation overhead rather than trainability constraints. Table 7 summarizes the results: across all four sequence lengths, EAGLE consistently achieves higher MFU and tokens-per-second throughput than Full Recompute, with throughput gains of 13.2%, 15.0%, 16.3%, and 17.9% at 4k, 8k, 16k, and 32k tokens respectively, while matching the Alloc/Peak memory footprint of selective recomputation. The improvement grows monotonically with sequence length, consistent with the fact that the recomputation cost of FlashAttention scales as $O(s^2)$ in the number of tokens and increasingly dominates the backward step, whereas the remaining components scale approximately linearly in $s$; as a result, EAGLE removes a larger fraction of recomputation overhead at longer contexts.

## E.2   GRADIENT CONSISTENCY WITH STANDARD AUTOGRAD

We verify that EAGLE's analytical-gradient implementation produces numerically equivalent gradients to standard automatic differentiation. On a single-GPU toy model comprising a MoE–MLA–MoE stack (hidden size 2048, 16 attention heads, FFN size 10944, MoE FFN size 1408, 8 experts

Table 7: Sequence length sensitivity on LLaMA3-8B comparing recomputation strategies.

| Seq. Length | Method | MFU | Tokens/s | Alloc Mem (GB) | Peak Mem (GB) |
|---|---|---|---|---|---|
| 4096 | No Recompute | 0.24 | 1481.10 | 27.45 | 34.20 |
| | Full Recompute | 0.18 | 1113.93 | 23.45 | 29.94 |
| | Selective Recompute | 0.19 | 1168.85 | 23.95 | 30.47 |
| | EAGLE | **0.21** | **1261.03** | **23.95** | **30.47** |
| | *EAGLE vs. Full* | **+16.7%** | **+13.2%** | – | – |
| 8192 | No Recompute | 0.26 | 1394.70 | 32.45 | 39.52 |
| | Full Recompute | 0.19 | 1043.26 | 24.45 | 31.01 |
| | Selective Recompute | 0.20 | 1090.37 | 25.45 | 32.07 |
| | EAGLE | **0.22** | **1199.86** | **25.45** | **32.07** |
| | *EAGLE vs. Full* | **+15.8%** | **+15.0%** | – | – |
| 16384 | No Recompute | 0.29 | 1298.40 | 42.46 | 50.17 |
| | Full Recompute | 0.22 | 970.78 | 26.45 | 33.14 |
| | Selective Recompute | 0.23 | 1011.18 | 28.45 | 35.26 |
| | EAGLE | **0.26** | **1128.67** | **28.45** | **35.26** |
| | *EAGLE vs. Full* | **+18.2%** | **+16.3%** | – | – |
| 32768 | No Recompute | 0.34 | 1092.07 | 62.47 | 71.46 |
| | Full Recompute | 0.25 | 823.14 | 30.45 | 37.40 |
| | Selective Recompute | 0.26 | 851.95 | 34.45 | 41.65 |
| | EAGLE | **0.30** | **970.58** | **34.45** | **41.65** |
| | *EAGLE vs. Full* | **+20.0%** | **+17.9%** | – | – |

Table 8: Gradient differences between EAGLE and FP32 Autograd on a MoE–MLA–MoE toy model.

| Setting | Precision | Infinity norm | Mean Abs. Error | Mean Rel. Error |
|---|---|---|---|---|
| Non-deterministic | BF16 | $1.22 \times 10^{-4}$ | $1.14 \times 10^{-7}$ | $5.24 \times 10^{-3}$ |
| | FP32 | $8.26 \times 10^{-7}$ | $5.99 \times 10^{-9}$ | $2.85 \times 10^{-4}$ |
| Fully deterministic | BF16 | 0 | 0 | 0 |
| | FP32 | 0 | 0 | 0 |

with top-6 routing and one shared expert), we run one forward–backward step with batch size 1 and sequence length 1024 under both EAGLE and a standard FP32 Autograd baseline. For every parameter tensor $T$, we compute three metrics between gradients $g_T^{\text{EAGLE}}$ and $g_T^{\text{AutoFP32}}$: the infinity norm $\|g_T^{\text{EAGLE}} - g_T^{\text{AutoFP32}}\|_\infty$, the mean absolute error $\frac{1}{n_T} \sum_i |g_{T,i}^{\text{EAGLE}} - g_{T,i}^{\text{AutoFP32}}|$, and the mean relative error $\frac{1}{n_T} \sum_i \frac{|g_{T,i}^{\text{EAGLE}} - g_{T,i}^{\text{AutoFP32}}|}{\max(|g_{T,i}^{\text{AutoFP32}}|, \epsilon)}$ with $\epsilon = 10^{-10}$. As shown in Table 8, non-deterministic kernels yield small discrepancies (e.g., $1.22 \times 10^{-4}$ in the infinity norm and $5.24 \times 10^{-3}$ in mean relative error for BF16), which are comparable to typical numerical noise in large-scale training, while fully deterministic settings drive all three metrics to zero in both BF16 and FP32. These results confirm that EAGLE and standard Autograd produce bit-wise identical gradients when sharing the same deterministic kernels, and that any residual differences under default settings are within the natural numerical variability of modern GPU training.

### E.3 CONVERGENCE BEHAVIOR AND RUN-TO-RUN VARIANCE

We further assess whether EAGLE affects convergence behavior or introduces additional training instability. First, we run five independent training runs of a 2-layer Transformer with MLA+MoE (16 experts, expert parallelism 8) at sequence length 4096 and global batch size 32 on a single A100 GPU. Each run trains for 10 steps, and we record the minimum iteration time per run. The timing statistics in Table 9 show coefficients of variation (standard deviation divided by mean) below 0.11% for all methods (No Recompute, Full Recompute, and EAGLE), indicating highly stable and reproducible performance, with EAGLE adding only $0.94\% \pm 0.14\%$ overhead relative to No Recompute versus $19.8\% \pm 0.19\%$ for Full Recompute. Second, we train an 8-layer model with global batch size

Table 9: Per-step iteration times (ms) for five independent runs of a 2-layer Transformer with different recomputation strategies.

| Run | No Recompute | Full Recompute | EAGLE |
|---|---|---|---|
| 1 | 2023.2 | 2422.4 | 2040.2 |
| 2 | 2026.1 | 2421.8 | 2044.0 |
| 3 | 2021.7 | 2425.2 | 2040.4 |
| 4 | 2025.9 | 2425.9 | 2043.4 |
| 5 | 2020.8 | 2426.2 | 2044.9 |
| Mean $\pm$ Std | $2023.54 \pm 2.15$ | $2424.30 \pm 1.84$ | $2042.58 \pm 1.92$ |
| CV (%) | 0.106 | 0.076 | 0.094 |
| Relative Overhead (%) | – | $19.8 \pm 0.19$ | $0.94 \pm 0.14$ |

Table 10: Mean absolute relative error (MARE) between per-step loss curves for different training configurations.

| Comparison | MARE |
|---|---|
| No Recompute vs. Full Recompute | $5.7 \times 10^{-5}$ |
| No Recompute vs. EAGLE | $8.9 \times 10^{-5}$ |
| No Recompute (run 1) vs. No Recompute (run 2) | $9.2 \times 10^{-5}$ |

16 on 4 GPUs for 200 steps and compute the mean absolute relative error (MARE) between per-step losses across different pairs of runs. As summarized in Table 10, the MARE between No Recompute and EAGLE ($8.9 \times 10^{-5}$) is of the same order as the natural run-to-run variability between two No Recompute runs ($9.2 \times 10^{-5}$), demonstrating that EAGLE does not introduce measurable changes to convergence trajectories or final model quality compared to standard recomputation.

### E.4 RELATIONSHIP TO PRIOR CHECKPOINTING AND SCHEDULING METHODS

EAGLE is complementary to prior work that optimizes where and when recomputation occurs. CheckMate's tensor rematerialization framework, for example, focuses on deriving optimal recomputation schedules that minimize peak memory for a given computational budget, whereas EAGLE targets the execution of each recomputation step by replacing redundant forward passes with analytical gradients and FlashAttention-based backward computation. In principle, EAGLE's kernels can be incorporated into CheckMate-style schedules so that each selected recompute region is executed more efficiently. Similarly, Adacc dynamically chooses recomputation locations and employs tensor compression to trade a small amount of accuracy for reduced activation memory, while EAGLE is strictly lossless and uses a static recomputation design based on architectural building blocks such as RMSNorm+Linear, MLP, MoE, and attention modules. Once Adacc has identified recomputation regions, EAGLE's analytical-gradient kernels can be applied to regions that terminate in linear or FlashAttention blocks, providing additional speedups on top of Adacc's memory savings.

