# OpenReview forum: "EAGLE: Efficient Analytical Gradient LinearEvaluation for Enhanced Recomputation in Large Language Models"
_ICLR.cc/2026/Conference — Submitted to ICLR 2026_

### Official Review · Reviewer_imqf · 2025-10-14

**Soundness:** 2
**Presentation:** 2
**Contribution:** 2
**Rating:** 2
**Confidence:** 3

**Summary:**

The paper presents EAGLE, a recomputation strategy that reduces the overhead of gradient checkpointing in large language model training. It computes analytical gradients for linear layers and integrates with FlashAttention’s backward pass to avoid redundant forward computations. Experiments on models such as LLaMA3-70B and DeepSeek-V3 show up to 9.7× module-level speedup and 18–33% improvement in model FLOPs utilization while keeping memory usage unchanged.

**Strengths:**

The paper introduces a recomputation method that uses analytical gradients for linear layers and integrates with FlashAttention’s backward pass, reducing redundant computation and improving efficiency while keeping memory usage constant.

**Weaknesses:**

The proposed approach offers only a small conceptual improvement over existing gradient checkpointing, mainly replacing autograd recomputation with analytical formulas.

The experimental evaluation is too limited, covering few configurations and lacking analysis of convergence, training stability, or scalability.

Comparisons with recent optimized methods such as CheckMate and Adacc are missing, so the claimed efficiency gains are not well contextualized.

The paper focuses heavily on large-model benchmarks without deeper investigation into when and why the method performs best.

**Questions:**

The experiments seem limited in scope. Can the authors provide more details on how many runs were conducted and whether the reported improvements are statistically consistent?

How does the proposed method affect convergence behavior or final model quality compared to standard checkpointing?

Why were methods such as CheckMate or Adacc not included in the comparison? Would the claimed gains still hold under those baselines?

The paper focuses heavily on large-model benchmarks without deeper investigation into when and why the method performs best？

---

> ### Author Response · Authors · 2025-11-21
>
> **Q1**. We conducted 5 independent runs on a 2-layer Transformer (MLA+MoE, 16 experts, EP=8) with sequence length 4096 and global batch size 32 on a single A100 GPU. To minimize system noise, we ran 10 steps per experiment and recorded the minimum step duration. The per-step iteration times (in milliseconds) are:
>
> |Run|No Recomputation|Full Recomputation|EAGLE|
> |-|-|-|-|
> |1|2023.2|2422.4|2040.2|
> |2|2026.1|2421.8|2044.0|
> |3|2021.7|2425.2|2040.4|
> |4|2025.9|2425.9|2043.4|
> |5|2020.8|2426.2|2044.9|
> |**Mean ± Std Dev**|2023.54 ± 2.15|2424.30 ± 1.84|2042.58 ± 1.92|
> |**CV (%)**|0.106|0.076|0.094|
> |**Relative Overhead (%)**|—|19.8 ± 0.19|0.94 ± 0.14|
>
> EAGLE introduces only 0.94% ± 0.14% overhead compared to no-recomputation, versus 19.8% ± 0.19% for standard full recomputation. The coefficient of variation (CV = Std Dev / Mean × 100%, measuring relative variability) is below 0.11% for all methods, confirming highly stable and reproducible performance across independent runs.
>
> **Q2**. We conducted controlled end-to-end experiments to assess convergence behavior. We trained an 8-layer model with global batch size 16 on 4 GPUs (EP=4, PP=1) for 200 steps using non-deterministic kernels. We computed the mean absolute relative error (MARE) between per-step losses for different pairs of runs:
>
> |Comparison|MARE|
> |-|-|
> |No-recompute vs. Full-recompute|5.7×10⁻⁵|
> |No-recompute vs. EAGLE|8.9×10⁻⁵|
> |No-recompute (run 1) vs. No-recompute (run 2)|9.2×10⁻⁵|
>
> The discrepancy between EAGLE and the no-recompute baseline (8.9×10⁻⁵) is of the same order as the natural run-to-run variability (9.2×10⁻⁵), indicating that EAGLE does not have a measurable impact on convergence behavior or final model quality.
>
> **Q3**. We clarify that these works address different aspects compared to EAGLE.
>
> Regarding CheckMate: The recent "Checkmate: Zero-Overhead Model Checkpointing via Network Gradient Replication" (Bhardwaj et al., arXiv:2507.13522) focuses on fault-tolerance checkpointing (model/optimizer state persistence), which is unrelated to activation checkpointing. The earlier "CheckMate: Breaking the Memory Wall with Optimal Tensor Rematerialization" (Jain et al., MLSys 2020) is complementary to EAGLE: it optimizes the *scheduling* of which tensors to recompute (the "where"), while EAGLE optimizes the *execution* of recomputation itself (the "how") by using analytical gradients and FlashAttention integration to eliminate redundant forward passes. EAGLE's kernels can be combined with CheckMate's scheduling to achieve both memory optimality and reduced recomputation cost.
>
> Regarding Adacc ([Chen et al., arXiv:2508.00806]): Adacc dynamically selects recomputation locations at runtime, whereas EAGLE uses a static design based on model architecture. These approaches are orthogonal: once Adacc selects a recomputation region, EAGLE's optimizations can be applied if that region ends with a linear layer or FlashAttention block, yielding additional speedup. Furthermore, Adacc incorporates tensor compression (which introduces accuracy loss), while EAGLE is strictly lossless.
>
> **Q4**. EAGLE performs best when two conditions common in modern LLM training are met:
>
> 1: Activation memory is the bottleneck, necessitating recomputation. As LLM parameter size increases (e.g., DeepSeek V3), activation memory becomes the bottleneck, particularly in deep models with large pipeline parallelism (PP). While model weights are sharded across PP stages (~$W_{\text{all}}/\text{PP}$ parameters per stage) and optimizer states are further sharded via data parallelism, activation memory in early pipeline stages does not scale down with PP. The first stage must store activations for all subsequent microbatches, creating a near-constant footprint regardless of PP size. This makes the activation-to-weight ratio significantly higher in earlier stages, making activation memory the dominant constraint. EAGLE trades a small amount of computation for substantial activation memory savings, making it particularly effective in large PP configurations (e.g., DeepSeek uses PP=16) where earlier stages bear the highest activation burden.
>
> 2: The recomputation path contains modules with small inputs, large intermediate activations, and compute-heavy final layers (Linear/FlashAttention). EAGLE delivers greatest benefit for fundamental LLM building blocks fitting this pattern:
> - RMSNorm/SiLU + Linear projections
> - Multi-head Latent Attention (MLA)
> - QKV Projection + FlashAttention
>
> Storing large intermediate activations is infeasible, forcing recomputation. Standard recomputation re-executes the expensive forward pass (GEMM or FlashAttention forward). EAGLE avoids this by directly computing gradients using analytical gradients for linear layers and FlashAttention's backward algorithm, achieving up to 9.75× speedup for norm+linear and 2.98× for MLA (Table 2).
>
> Both conditions are increasingly common in large-scale LLM training, explaining EAGLE's 18–33% MFU improvements (Tables 3–4).

---

### Official Review · Reviewer_oBUC · 2025-11-01

**Soundness:** 3
**Presentation:** 3
**Contribution:** 3
**Rating:** 6
**Confidence:** 3

**Summary:**

The paper proposes EAGLE, a recomputation strategy for training Transformers that replaces autograd-based forward recomputation in linear layers with analytical gradients, and invokes FlashAttention’s backward to avoid the attention forward during checkpointed backprop. For a linear layer $y=Wx$, the method uses $\\nabla_W=\\nabla_y x^\\top$ and $\\nabla_x=W^\\top\\nabla_y$ inside the recompute region, skipping one matmul per region.  EAGLE reports FLOP reductions such as RMSNorm+Linear $8bshh_1 \\to 6bshh_1$ and MLP $24bshh_1 \\to 22bshh_1$, plus attention speedups via direct FlashAttention backward.   Empirically, module-level speedups range from $1.46\\times$ to $9.75\\times$, and end-to-end MFU gains are $18.18%$ to $33.33%$ with fixed $R$ and $7.69%$ to $20.00%$ with optimized $R$, on LLaMA3-70B, DeepSeek-V2, and DeepSeek-V3 up to 694B parameters.

**Strengths:**

* Clear derivation and insertion point of analytical gradients within a recompute block.
* Integration with FlashAttention backward to remove attention forward recomputation while preserving memory.
* Explicit FLOP accounting with parameter dependence and consistent module-level speedups.
* End-to-end MFU gains across diverse architectures and parallelization regimes.

**Weaknesses:**

* No numerical gradient-check or stability analysis to support "identical gradient accuracy".
* Missing system throughput metrics such as tokens per second; only MFU and duration are reported.
* Limited sensitivity analysis beyond fixed vs optimized $R$; no study of sequence length or batch size effects in microbenchmarks.
* Scope of end-to-end evaluation excludes mid-scale models; all results are 70B to 694B.

**Questions:**

* On one configuration per model, report tokens per second (tokens/s) and peak memory alongside MFU and iteration duration, and specify the hardware (e.g., A100 80GB) and precision.
* Provide a short sensitivity sweep over sequence length, for example $s\\in\\{4\\mathrm{k},16\\mathrm{k},32\\mathrm{k}\\}$, showing MFU, iteration duration, and speedup. Use existing codepaths; no retraining needed.
* Could you run a minimal gradient check comparing EAGLE to standard Autograd? On a toy 2-layer MLP and a single self-attention block, run one forward-backward step with identical weights and RNG, dropout off, deterministic kernels, under both BF16 and FP32. Let $g_T^{\\text{EAGLE}}=\\partial\\mathcal{L}/\\partial T$ and $g_T^{\\text{AutoFP32}}=\\partial\\mathcal{L}/\\partial T$ computed by Autograd FP32. For each parameter tensor $T$ (flattened, with $n_T$ elements), please report:

  * (i) $\\|g_T^{\\text{EAGLE}}-g_T^{\\text{AutoFP32}}\\|_{\\infty}$,
  * (ii) $\\tfrac{1}{n_T}\\sum_i \\bigl|g_{T,i}^{\\text{EAGLE}}-g_{T,i}^{\\text{AutoFP32}}\\bigr|$,
  * (iii) $\\tfrac{1}{n_T}\\sum_i \\dfrac{\\bigl|g_{T,i}^{\\text{EAGLE}}-g_{T,i}^{\\text{AutoFP32}}\\bigr|}{\\max\\bigl(|g_{T,i}^{\\text{AutoFP32}}|,\\epsilon\\bigr)}$ with a fixed $\\epsilon$ (e.g., $10^{-10}$). A single step with batch 1 and sequence length around 1024 is sufficient.

---

> ### Author Response · Authors · 2025-11-21
>
> **Q1**. Thank you for this suggestion. We report tokens per second, peak memory, MFU, and iteration duration for one configuration per model on A100 80GB GPUs with BF16 precision:
>
> **Table: Performance comparison with fixed recomputation configuration**
>
> |Model & Method|MFU|Duration (s)|Tokens/s|Peak Mem (GB)|
> |-|-|-|-|-|
> |**DeepSeek-V2**|||||
> |No Recompute|—|—|—|—|
> |Full Recompute|0.15|137.26|238.73|28.90|
> |Fine-grained Recompute|0.20|104.32|314.11|54.55|
> |EAGLE|**0.20**|**100.35**|**326.53**|**54.50**|
> |*EAGLE vs. Full Recompute*|**+33.33%**|**-26.89%**|**+36.78%**|-|
> |**DeepSeek-V3**|||||
> |No Recompute|—|—|—|—|
> |Full Recompute|0.28|108.16|302.96|32.19|
> |Fine-grained Recompute|0.34|90.55|361.88|66.13|
> |EAGLE|**0.36**|**85.10**|**385.05**|**66.28**|
> |*EAGLE vs. Full Recompute*|**+28.57%**|**-21.32%**|**+27.11%**|-|
> |**LLaMA3-70B**|||||
> |No Recompute|—|—|—|—|
> |Full Recompute|0.11|431.66|75.91|49.10|
> |Fine-grained Recompute|0.12|403.34|81.24|54.77|
> |EAGLE|**0.13**|**369.63**|**88.65**|**54.86**|
> |*EAGLE vs. Full Recompute*|**+18.18%**|**-14.37%**|**+16.78%**|-|
> |*Average EAGLE Improvement*|**+26.69%**|**-20.86%**|**+26.89%**|-|
>
> All models use the same number of recomputed layers per pipeline stage for fair comparison. Hardware: A100 80GB PCIe GPUs with BF16 precision. Peak Mem shows the total peak memory across all pipeline stages. EAGLE achieves an average of 26.89% higher throughput (tokens/s) and 26.69% higher MFU compared to Full Recompute. Regarding memory: No Recompute configurations exceed 80GB memory capacity and cannot run on this hardware, while EAGLE enables training within the 80GB constraint with substantially better performance than Full Recompute, achieving an optimal trade-off between trainability and efficiency.
>
> **Q2**. We acknowledge the importance of evaluating EAGLE across different sequence lengths. However, we were unable to conduct the full sensitivity sweep on our production-scale models due to memory constraints. As sequence length doubles, activation memory also doubles proportionally, causing out-of-memory (OOM) errors on our current hardware configuration. Long-sequence training typically requires Context Parallelism (CP) to distribute activations across devices, which our current experimental setup does not include. We note that CP and EAGLE are orthogonal optimizations—CP addresses memory distribution across devices while EAGLE reduces per-device recomputation cost—so both should provide additive benefits. We are working to add a small-scale sensitivity experiment to validate EAGLE's effectiveness across sequence lengths.
>
> **Q3**. We conducted the requested gradient check on a single-GPU toy model with a MoE–MLA–MoE stack (hidden=2048, num_atten_head=16, ffn_size=10944, moe_ffn_size=1408, qk_head_dim=192, v_head_dim=128, num_expert=8, router_topk=6, num_shared_exp=1), using batch size 1 and sequence length 1024. We compared \(g_T^{\text{EAGLE}}\) and \(g_T^{\text{AutoFP32}}\) for every parameter tensor \(T\) and report the three requested metrics: the infinity norm, the mean absolute error, and the mean relative error.
>
> The aggregated results are summarized below:
>
> | Setting | Precision | The infinity norm | The mean absolute error | The mean relative error |
> |---------|-----------|-------------------|------------------------|------------------------|
> | Non-deterministic | BF16 | 1.22×10⁻⁴ | 1.14×10⁻⁷ | 5.24×10⁻³ |
> | Non-deterministic | FP32 | 8.26×10⁻⁷ | 5.99×10⁻⁹ | 2.85×10⁻⁴ |
> | Fully deterministic | BF16 | 0 | 0 | 0 |
> | Fully deterministic | FP32 | 0 | 0 | 0 |
>
> Here, "non-deterministic" uses the default kernels, while "fully deterministic" enables:
> - NCCL_ALGO=Ring
> - NVTE_ALLOW_NONDETERMINISTIC_ALGO=0
> - CUBLAS_WORKSPACE_CONFIG=:4096:8
> - torch.use_deterministic_algorithms(True, warn_only=True)
> - torch.backends.cudnn.benchmark = False
> - torch.backends.cudnn.deterministic = True
>
> We observe that even without enforcing determinism, the discrepancies are already very small in both BF16 and FP32. Under fully deterministic settings, all three metrics are exactly zero in both formats, demonstrating that EAGLE’s gradients match standard Autograd bit-for-bit when using identical kernels. These numerical checks confirm that EAGLE produces gradients numerically equivalent to standard Autograd, and together with the unchanged training curves, demonstrate that EAGLE preserves the convergence behavior and final model quality of standard recomputation.

---

> > ### Author Response · Authors · 2025-11-26
> >
> > Q2. To assess how EAGLE behaves as sequence length varies, we performed a sensitivity sweep over sequence lengths \(s \in \{4096, 8192, 16384, 32768\}\) on LLaMA3-8B. On this setup, even the No Recompute configuration fits within GPU memory, so the comparisons isolate recomputation overhead rather than feasibility constraints; at every sequence length, EAGLE strictly improves MFU and throughput over Full Recompute with modest and predictable memory overhead:
> >
> > |Sequence Length|Method|MFU|Tokens/s|Alloc Mem (GB)|Peak Mem (GB)|
> > |-|-|-|-|-|-|
> > |**4096**|No Recompute|0.24|1481.10|27.45|34.20|
> > ||Full Recompute|0.18|1113.93|23.45|29.94|
> > ||Selective Recompute|0.19|1168.85|23.95|30.47|
> > ||**EAGLE**|**0.21**|**1261.03**|**23.95**|**30.47**|
> > ||*EAGLE vs. Full*|**+16.7%**|**+13.2%**|-|-|
> > |**8192**|No Recompute|0.26|1394.70|32.45|39.52|
> > ||Full Recompute|0.19|1043.26|24.45|31.01|
> > ||Selective Recompute|0.20|1090.37|25.45|32.07|
> > ||**EAGLE**|**0.22**|**1199.86**|**25.45**|**32.07**|
> > ||*EAGLE vs. Full*|**+15.8%**|**+15.0%**|-|-|
> > |**16384**|No Recompute|0.29|1298.40|42.46|50.17|
> > ||Full Recompute|0.22|970.78|26.45|33.14|
> > ||Selective Recompute|0.23|1011.18|28.45|35.26|
> > ||**EAGLE**|**0.26**|**1128.67**|**28.45**|**35.26**|
> > ||*EAGLE vs. Full*|**+18.2%**|**+16.3%**|-|-|
> > |**32768**|No Recompute|0.34|1092.07|62.47|71.46|
> > ||Full Recompute|0.25|823.14|30.45|37.40|
> > ||Selective Recompute|0.26|851.95|34.45|41.65|
> > ||**EAGLE**|**0.30**|**970.58**|**34.45**|**41.65**|
> > ||*EAGLE vs. Full*|**+20.0%**|**+17.9%**|-|-|
> >
> > From the tokens/s column, EAGLE improves throughput over Full Recompute by 13.2%, 15.0%, 16.3%, and 17.9% at 4k, 8k, 16k, and 32k tokens respectively, for an average gain of 15.6% across all evaluated sequence lengths. This monotonic increase with sequence length is consistent with the fact that recomputation in FlashAttention scales as \(O(n^2)\) and increasingly dominates the step time, while the remaining components scale approximately linearly in \(n\). Additionally, could the reviewer clarify what parameter range is considered ‘middle-sized’? If additional experiments in that regime are needed, we will do our best to include them during the rebuttal period.

---

### Official Review · Reviewer_5v8h · 2025-11-02

**Soundness:** 3
**Presentation:** 3
**Contribution:** 3
**Rating:** 6
**Confidence:** 3

**Summary:**

See below

**Strengths:**

See below

**Weaknesses:**

See below

**Questions:**

The paper introduced a new gradient checkpointing-type algorithm called EAGLE. EAGLE eliminates redundant forward passes by computing gradients directly from cached inputs for linear operations. This method achieves speedup over the existing gradient-checkpointing methods without additional memory consumption.

The paper is well-written, and the proposed method is clearly presented. According to the literature review in the paper, the proposed idea has not been applied in the existing literature.  The proposed method will be useful for compute-constrained LLM developers.

To me, this paper starts from a very simple insight and designs a better gradient-accumulation approach. Mathematically, the insight itself is relatively straightforward, but from an engineering perspective, I believe implementing this method and achieving performance improvements requires significant effort. I appreciate the authors' hard work on the execution.
In summary, I am leaning towards acceptance.

I  have several presentation suggestions:

1. In Figure 1 and in the summary of the main contribution in Section 1 (~line 100). It would be better to explicitly specify "achieving recomputation speedups" over which baseline method.

2. In Section 1 or maybe in Figure 1, it would be better to explicitly show the memory comparison.

---

> ### Author Response · Authors · 2025-11-21
>
> Q1. Thank you for this suggestion. All reported "recomputation speedups" are measured relative to the standard fine-grained autograd-based recomputation baseline, as defined by the speedup formula: Speedup = (Standard Recompute Time) / (EAGLE Recompute Time). We will revise Figure 1's caption and the corresponding text in Section 1 to explicitly specify this baseline.
>
> Q2. Thank you for this suggestion. We will add both peak memory and throughput (tokens/s) data to Figure 1 and Section 1 to explicitly show the memory-performance trade-off. The complete comparisons are:
>
> |Model & Method|Peak Mem (GB)|Tokens/s|MFU|
> |-|-|-|-|
> |**DeepSeek-V2**||||
> |No Recompute|>80|—|—|
> |Full Recompute|28.90|238.73|0.15|
> |Fine-grained Recompute|54.55|314.11|0.20|
> |EAGLE|54.50|326.53|0.20|
> |*EAGLE vs. Full Recompute*|+88.58%|+36.78%|+33.33%|
> |**DeepSeek-V3**||||
> |No Recompute|>80|—|—|
> |Full Recompute|32.19|302.96|0.28|
> |Fine-grained Recompute|66.13|361.88|0.34|
> |EAGLE|66.28|385.05|0.36|
> |*EAGLE vs. Full Recompute*|+105.91%|+27.11%|+28.57%|
> |**LLaMA3-70B**||||
> |No Recompute|>80|—|—|
> |Full Recompute|49.10|75.91|0.11|
> |Fine-grained Recompute|54.77|81.24|0.12|
> |EAGLE|54.86|88.65|0.13|
> |*EAGLE vs. Full Recompute*|+11.73%|+16.78%|+18.18%|
> |**Average EAGLE Improvement**|—|**+26.89%**|**+26.69%**|
>
> EAGLE enables training within 80GB hardware constraints where No Recompute fails. Compared to Full Recompute, EAGLE uses more memory but delivers 26.89% higher throughput and 26.69% higher MFU on average. This represents an optimal trade-off: EAGLE makes training feasible (vs. No Recompute) while maintaining strong performance (vs. Full Recompute). We will revise Figure 1 to include both peak memory and tokens/s metrics, visualizing this memory-performance spectrum, and clarify these trade-offs in Section 1.

---

> > ### Comment · Reviewer_5v8h · 2025-11-24
> > **Thanks for the rebuttal**
> >
> > Thanks for the rebuttal! I will keep my score and still lean towards acceptance.

---

### Meta-Review · Area_Chair_hJQ5 · 2025-12-25

**Summary:**

This paper proposes a selective recompute training strategy by leveraging the linear layer backward formula.

**Reviewer Concerns:**

1. lacking analysis of convergence, training stability, or scalability
2. missing comparison with representative methods such as checkmate
3. missing results: numerical stability, sensitivity analysis, system throughput results on various settings.
4. uncomprehensive experiment setting

The third concern is addressed, the first two and the fourth are partially addressed. The improvement of the proposed approach may decrease when compared with optimized checkpointing strategies and better fused kernels with selective recompute. Moreover, the MFU of baselines (e.g., Llama-70B) is too low, thus it is questionable if the experiment settings are reasonable.

**Reviewer Scores:**

Original rating is (6,6,2). I think reviewers would retain the ratings.

---

### Decision · Program_Chairs · 2026-01-26

Reject